# Crystallization and Crystallographic Analysis of a *Bradyrhizobium Elkanii* USDA94 Haloalkane Dehalogenase Variant with an Eliminated Halide-Binding Site

Tatyana Prudnikova [1,2,†], Barbora Kascakova [1,†], Jeroen R. Mesters [3] ⬤, Pavel Grinkevich [1], Petra Havlickova [1], Andrii Mazur [1,2], Anastasiia Shaposhnikova [1,2] ⬤, Radka Chaloupkova [4], Jiri Damborsky [4,5] ⬤, Michal Kuty [1,2] and Ivana Kuta Smatanova [1,2,*] ⬤

1 Faculty of Science, University of South Bohemia in Ceske Budejovice, Branisovska 1760, 37005 Ceske Budejovice, Czech Republic

2 Center of Nanobiology and Structural Biology, Institute of Microbiology of the Czech Academy of Sciences, Zamek 136, 37333 Nove Hrady, Czech Republic

3 Institute of Biochemistry, University of Lübeck, Ratzeburger Allee 160, 23538 Lübeck, Germany

4 Loschmidt Laboratories, Department of Experimental Biology and RECETOX, Faculty of Science, Masaryk University, Kamenice 5/A4, 62500 Brno, Czech Republic

5 International Clinical Research Center, St. Anne's University Hospital Brno, Pekarska 53, 65691 Brno, Czech Republic

* Correspondence: ivanaks@seznam.cz or talianensis@gmail.com

† These authors contributed equally to this work.

**Abstract:** Haloalkane dehalogenases are a very important class of microbial enzymes for environmental detoxification of halogenated pollutants, for biocatalysis, biosensing and molecular tagging. The double mutant (Ile44Leu + Gln102His) of the haloalkane dehalogenase DbeA from *Bradyhizobium elkanii* USDA94 (DbeAΔCl) was constructed to study the role of the second halide-binding site previously discovered in the wild-type structure. The variant is less active, less stable in the presence of chloride ions and exhibits significantly altered substrate specificity when compared with the DbeAwt. DbeAΔCl was crystallized using the sitting-drop vapour-diffusion procedure with further optimization by the random microseeding technique. The crystal structure of the DbeAΔCl has been determined and refined to the 1.4 Å resolution. The DbeAΔCl crystals belong to monoclinic space group *C*121. The DbeAΔCl molecular structure was characterized and compared with five known haloalkane dehalogenases selected from the Protein Data Bank.

**Keywords:** Haloalkane dehalogenase; halide-binding site; random microseeding

## 1. Introduction

Hazardous halogenated compounds are an important class of environmental pollutants. An obvious critical step in the potential biodegradation pathway is the dehalogenation process [1,2]. Haloalkane dehalogenases (HLDs) play an essential role in biodegradation of the halogenated pollutants. HLDs are predominantly bacterial enzymes that belong to the superfamily of α/β-hydrolases and catalyze the hydrolytic conversion of a wide range of halogenated aliphatic compounds, and therefore play an important role in bioremediation [3] and industrial biocatalytic processes [4]. An aliphatic alcohol, a halide and a hydrogen cation are released during the enzymatic dehalogenation of haloalkanes by HLDs. The tertiary structures of HLDs are composed of a conserved α/β-hydrolase core domain and an α-helical cap domain [5]. The core domain is responsible for the catalytic reaction of the enzyme

and the cap domain is essential for substrate specificity and recognition [6]. A deep cleft is situated between these two domains, allowing the solvent to access the buried active site. The active site is composed of two halide-anion stabilizing residues and the catalytic triad consisting of a nucleophile, a base and an acid [7]. HLDs can be divided into three subfamilies, HLD-I, HLD-II and HLD-III, according to the composition of the catalytic residues and the anatomy of the cap domain [4].

A novel HLD DbeA from *B. elkanii* USDA94, a member of HLD-II subfamily [4], was structurally and biochemically characterized [7]. The structure of DbeA wild type was determined to 2.2 Å resolution and displays a typical topology of the α/β-hydrolases (EC 3.8.1.5). The unique feature of the DbeA structure is the presence of two halide-binding sites, both fully occupied by chloride anions [7]. The first halide-binding site is located in the protein active site and is involved in substrate binding and stabilization of halogen ion produced during dehalogenation reaction. DbeA active site consists of five catalytic residues: two halide stabilizing residues (Trp104 and Asn38) and three amino acids essential for the catalytic activity of the enzyme [2,4]: the nucleophile Asp103, the catalytic base His271, and the catalytic acid Glu127. The second halide-binding site in DbeA is unique and has never been observed within HLD structures deposited in the PDB [8]. The second halide-binding site, which is buried in the protein core domain and located approximately 10 Å far from the first halide-binding site, is formed by five amino-acid residues: Ile44, Gln274, Gln102, Gly37 and Thr40 [7]. Superposition of the DbeA structure with other related HLD-II members revealed the presence of two unique amino acids in the second halide-binding site: Gln102 instead of a typical His and Ile44 as a substitution of an ordinary Leu, thereby sufficiently increasing the cavity volume to accommodate the second halide ion. The variant DbeAΔCl (Ile44Leu+Gln102His) was constructed and biochemically characterized to elucidate the role of the second halide-binding site in structure and function of DbeA [7].

Removal of the second halide binding site in DbeA significantly changed the substrate specificity of DbeAΔCl and reduced the catalytic activity by an order of magnitude towards most of the tested substrates [7]. Wild-type DbeA is more active, its melting temperature rises with an increasing concentration of chloride salts and the binding energy for chloride ions is higher when compared with the DbeAΔCl variant [7]. It was suggested that the chloride anion bound in a vicinity of second binding site may increase basicity of catalytic histidine and consequently accelerate the nucleophilic addition of water to the alkyl-enzyme intermediate [7]. In previous attempts, the crystallization of DbeAΔCl was unsuccessful. The obtained crystals were very unstable, sensitive to mechanical stress and poorly diffracted X-rays to about 10 Å resolution. It took several years to grow crystals with an improved diffraction quality. Here, we report the successful crystallization, structure determination and further characterization of DbeAΔCl variant.

## 2. Materials and Methods

### 2.1. Gene Synthesis, Cloning, Expression and Protein Purification

The recombinant gene dbeAΔCl-His6 (Ile44Leu + Gln102His) was synthesized artificially (Entelechon, Regensburg, Germany) according to the DbeA sequence [7] (Table 1). The restriction endonucleases NdeI and XhoI (Fermentas, Burlington, Canada) and T4 DNA ligase (Promega, Madison, USA) were applied to transfer the synthesized gene into the expression vector pET-21b (Novagen, Madison, USA). In order to overexpress DbeAΔCl in *E. coli* BL21(DE3) cells, the final genes were transcribed by T7 RNA polymerase, which is expressed by the isopropyl β-D-1-thiogalactopyranoside (IPTG)-inducible lac UV5 promoter. Cells containing the plasmid were cultured in Luria broth medium at 310 K. When the culture reached an optical density of 0.6 at a wavelength of 600 nm, gene expression (at 293 K) was induced by the addition of 0.5 mM IPTG. The cells were subsequently harvested and disrupted by sonication using a Soniprep 150 (Sanyo Gallenkamp PLC, Loughborough, England). The supernatant was collected after centrifugation at 100,000 g for 1 h. The crude extract was further purified on a HiTrap Chelating HP 5 ml column charged with $Ni^{2+}$ ions (GE Healthcare, Uppsala, Sweden). The His-tagged enzyme was bound to the resin in the presence of 20 mM potassium phosphate buffer

pH 7.5, 0.5 M sodium chloride, 10 mM imidazole. Unbound and non-specifically bound proteins were washed out by a buffer containing 37.5 mM imidazole. The target enzyme was eluted with a buffer containing 300 mM imidazole. The active fractions were pooled and dialyzed overnight against 50 mM Tris-HCl pH 7.5. The DbeAΔCl enzyme was stored at 277 K in 50 mM Tris-HCl pH 7.5 buffer prior to analysis. The DbeAΔCl production information is summarized in Table 1.

**Table 1.** Production specifics for DbeAΔCl.

| Source Organism | *Bradyrhizobium Elkanii* USDA94 |
|---|---|
| DNA source | Artificially synthesized DNA |
| Transport vector | pMA |
| Expression vector | pET-21b |
| Expression host | *E. coli* BL21(DE3) |
| Complete amino acid sequence of DbeAΔCl | MTISADISLHHRAVLGSTMAYRETGRSDAPHVLFLHGNPTSSYL WRNIMPLVAPVGHCIAPDLIGYGQSGKPDISYRFFDQADY LDALIDELGIASAYLVAHDWGTALAFHLAARRPQLVRGLA FMEFIRPMRDWSDFHQHDAARETFRKFRTPGVGEAMILDN NAFVERVLPGSILRTLSEEEMAAYRAPFATRESRMPTLML PRELPIAGEPADVTQALTAAHAALAASTYPKLLFVGSPGA LVSPAFAAEFAKTLKHCAVIQLGAGGHYLQEDHPEAIGRS VAGWIAGIEAASAQRHAALEHHHHHH |

## 2.2. Crystallization

The freshly isolated and purified DbeAΔCl protein was crystallized at a concentration of 30 mg.ml⁻¹ in 50 mM Tris–HCl buffer pH 7.5 by the sitting-drop vapour-diffusion procedure [9]. For initial screening several commercial precipitants kits were used: JBScreen Classic Kits № 1-10 and Wizard I-III (Jena Bioscience GmbH, Jena, Germany), Morpheus®HT-96, JCSG-plus™ HT-96, PACT premier™ HT-96 and Structure Screen 1 + 2 HT-96 kit (Molecular Dimensions Ltd (MDL), Suffolk, UK), Crystal Screen kit, PEGRx HT™ and PEG/Ion HT™ (Hampton Research (HR), Aliso Viejo, USA) and Axygen I-VIII crystallization kits (Axygen Biosciences, Union City, USA). The CombiClover 96 well plates (MDL, Suffolk, UK) for manual screening experiments as well as Swissi polystyrene MRC 2-drop plate (MDL, Suffolk, UK) were utilized for the initial screening on an Oryx3 robot (Douglas Instruments Ltd, Hungerford, UK) for the DbeAΔCl protein.

The hanging drop crystallization trials were carried out in Limbro 24 well plates (HR, Aliso Viejo, USA). The Douglas Instruments, USA Vapour Batch 96 well plates were used to perform the microbatch under oil crystallization [10]. Macroseeding experiments [11] were carried out by lowering the protein concentration to 20–25 mg.ml⁻¹. The counter-diffusion crystallization was performed in single glass capillaries with inner diameters ranging from 0.1 to 0.4 mm (HR, Aliso Viejo, USA) using a three-layer configuration [12].

## 2.3. Data Collection, Processing and Structure Solution

X-ray diffraction data at 100 K were collected to the 1.4 Å resolution at the BESSY II electron storage ring on beamline MX 14.1 of the Helmholtz-Zentrum Berlin (Berlin-Adlershof, Germany; [13]). 2500 images were processed with the graphical user interface XDSAPP [14] for running XDS [15]. Phasing by molecular replacement was performed using MOLREP [16] and the structure of DbeA as the template (PDB code 4k2a; [7]). One molecule was found in the asymmetric unit of DbeAΔCl. Structure refinement and model building was performed using isotropic and anisotropic refinement protocols in REFMAC5 [17] and Coot [18] from the CCP4 package [19], respectively. The quality of the protein models was confirmed with MolProbity [20,21] and wwPDB [22] validation servers. The structure of the DbeAΔCl has been deposited to the Protein Data Bank under accession code 6s42.

Figures with the structure were prepared using the program PyMOL [23]. The complete information about the data collection, processing, and refinement statistics are provided in Table 2.

**Table 2.** Data collection and crystallographic statistics.

| X-ray Diffraction Data Collection Statistics | |
|---|---|
| Space group | C121 |
| Cell parameters (Å, °) | a = 128.95, b = 63.95, c = 46.05; $\alpha = \gamma = 90$, $\beta = 106.27$ |
| Wavelength (Å) | 0.918 |
| Resolution (Å) | 1.39 |
| Number of unique reflections | 68,322 |
| Redundancy | 2.18 (2.20) |
| Completeness (%) | 96.13 (92.17) |
| $R_{merge}$ [#] | 4.9 (26.3) |
| Average $I/\sigma(I)$ | 12.72 (3.19) |
| Wilson B (Å$^2$) | 20.8 |
| **Refinement Statistics** | |
| Resolution range (Å) | 41.96–1.4 (1.43–1.39) |
| No. of reflections in working set | 64,907 (4,609) |
| R value (%) [##] | 13.98 |
| $R_{free}$ value (%) [###] | 15.22 |
| RMSD bond length (Å) | 0.006 |
| RMSD angle (°) | 1.635 |
| No. of atoms in AU | 2,823 |
| No. of protein atoms in AU | 2,333 |
| No. of water molecules in AU | 470 |
| No. of iodide ions in AU | 8 |
| No. of chloride ions in AU | 3 |
| Mean B value (Å$^2$) | 13.42 |
| **Ramachandran Plot Statistics** | |
| - Residues in favoured regions (%) | 97.2 |
| - Residues in allowed regions (%) | 100 |
| PDB code | 6s42 |

The data in parentheses refer to the highest-resolution shell. [#] $R_{merge} = (|I_{hkl} - \langle I \rangle|)/I_{hkl}$, where the average intensity $\langle I \rangle$ is taken over all symmetry equivalent measurements and $I_{hkl}$ is the measured intensity for any given reflection; [##] R-value $= ||F_o| - |F_c||/|F_o|$, where $F_o$ and $F_c$ are the observed and calculated structure factors, respectively; [###] $R_{free}$ is equivalent to R value but is calculated for 5% of the reflections chosen at random and omitted from the refinement process.

## 3. Results and Discussion

The crystallization procedure previously successfully used for growing of the DbeAwt protein [24] was applied to prepare crystals of freshly purified DbeAΔCl protein. Initially, only a light amorphous precipitation was observed. Further optimization of the crystallization conditions was carried out by variation of the protein and precipitant concentrations (PEG and salt) and drop protein-reservoir ratio composition. This optimization did not improve the results. Additional screening was conducted by applying further commercial crystallization kits: JBScreen Classic Kits № 1-10 and Wizard I-III (Jena Bioscience GmbH, Jena, Germany). Again, only very small and thin needle crystals or a heavy amorphous precipitation were observed. The next optimization step was based on moving the system closer to the metastable zone based on the phase transition diagram by decreasing the concentration of the crystallization drop components. The reduction of the DbeAΔCl concentration and variation of the precipitant concentrations yielded small microcrystals (Figure 1a) and a two-dimensional (2D) single needle crystals (Figure 1b) within a period of 1-3 weeks. The microcrystals were grown within three weeks from precipitant consisting of 28% (w/v) PEG 4000, 0.2 M Li$_2$SO$_4$ in 0.1 M Tris pH 8.2 buffer and a 30 mg.ml$^{-1}$ protein concentration. The small needle crystals were observed after 10 days at a 10 to

30 mg.ml$^{-1}$ protein concentration in precipitant composed of 12– 17% (w/v) PEG 3350, 115– 125 mM MgCl$_2$ in 100 mM Tris-HCl 7.5 buffer.

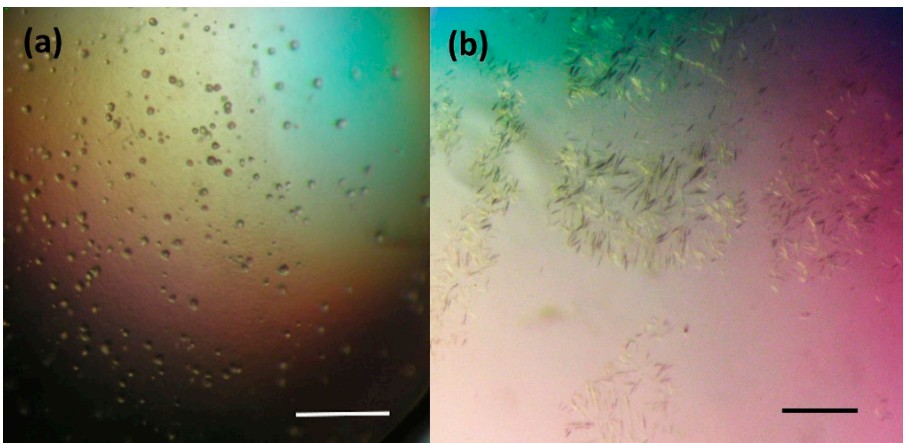

**Figure 1.** Results of initial crystallization experiments of DbeAΔCl protein from *B. elkanii* USDA94: (**a**) microcrystals and (**b**) small 2D needle crystals. The scale bar represents 100 μm.

In order to improve the crystal quality, Morpheus®HT-96, JCSG-plus™ HT-96, PACT premier™ HT-96 and Structure Screen 1 + 2 HT-96 kit (Molecular Dimensions Ltd (MDL), Suffolk, UK), Crystal Screen kit, PEGRx HT™ and PEG / Ion HT™ (Hampton Research (HR), Aliso Viejo, USA) and Axygen I-VIII crystallization kits (Axygen Biosciences, Union City, USA) were applied. Finally, small 3D crystals (Figure 2a) with a dimension of about 15 × 5 × 35 μm were grown from a solution containing 26.57% (w/v) hexanediol at 295 K over 10 days. The crystals diffracted X-rays to a maximum resolution of 8–10 Å. The quality of these crystals did not allow to record good diffraction data and additional strategies were needed to improve the size and shape of obtained crystals.

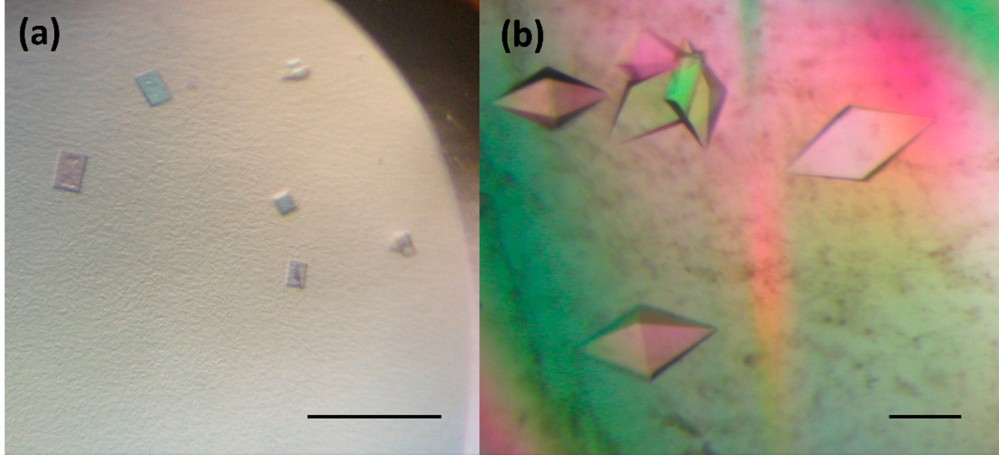

**Figure 2.** DbeAΔCl crystals used for diffraction analysis: (**a**) small 3D crystals grown at 26.5% (w/v) hexanediol and (**b**) big 3D crystals developed by random seeding experiments. The scale bar represents 100 μm.

Further optimization using the precipitant solution mentioned above was pursued by setting up the experiments at a lower temperature (193K), by application of the Additive Screen (HR) and by variation of the protein concentration from 10 to 40 mg.ml$^{-1}$, however, without success. The same conditions were tested in a hanging drop vapour-diffusion, microbatch under oil, counter-diffusion and macroseeding procedure. All these experiments did not significantly improve the diffraction quality of the crystals. Next, the small crystals (Figure 2a) were used for random microseeding experiments [25] with application of two commercial crystallization kits: PACT premier™ HT-96 and Structure Screen 1 + 2 HT-96 kit (MDL), which finally resulted in the appearance of 3D crystals with an average dimensions 70 × 75 × 100 μm within a period of two weeks (Figure 2b). These crystals, later used for X-ray data collection experiments, were obtained by lowering the protein concentration to 20 mg.ml$^{-1}$ and applying the following precipitant: 0.2M sodium iodide and 20% (w/v) PEG 3350.

Crystals of DbeAΔCl diffracted X-rays to 1.4 Å resolution and belong to the monoclinic base-centered space group *C*121. The diffraction data allowed localizing 294 amino-acid residues fitting to the one molecule in the asymmetric unit. The overall shape of DbeAΔCl is like a block with 2,823 non-hydrogen atoms (470 water molecules, 8 iodide anions, 3 chloride anions and hexanediol molecule) and corresponds to the canonical architecture of HLDs of the α/β hydrolase fold superfamily (Figure 3a). The structural organization of DbeAΔCl displays two compact domains: an α/β hydrolase core domain and a helical cap domain with the active site located between them. The cap domain (residues 134–214) consists of five α-helices (α4, α5¸ α5´, α6, α7 and α8) and six loops, together forming a lid, protecting the active site cavity. The core domain (residues 4–133 and 215–298) consists of a central twisted eight-stranded β-sheet with the β2 strand running antiparallel. The β-sheet region is flanked by six α-helices: two elements (α1 and α2) cover one side and the remaining four (α3, α9, α10 and α11) the other side [2,26] (Figure 3a). The protein displays a monomer as the biological unit according to the analysis of crystal contacts between molecules in the unit cell and crystal packing. The exploration of macromolecular interfaces by PDBePISA server [27] underpins the monomeric nature of the protein.

The DbeAΔCl active site displays a substrate-binding pocket typical for all haloalkane dehalogenases. The enzyme's active site cavity contains the catalytic triad consisting of Asp103, His 271 and Glu127. The nucleophile Asp103 is located at the turn between β-strand β5 and helix α3. The catalytic base His271 is positioned on the loop joining β8 and α11. The catalytic acid Glu127 is located behind β-strand β6. Inspecting the electron density map, one iodide anion and one molecule of hexanediol as components of precipitant cocktail were identified near the DbeAΔCl active site. The iodide ion is mainly stabilized by interactions with the N atoms of two halide-binding residues: Asn34 N$^{\delta 1}$ and Trp104 N$^{\epsilon 1}$ with distances of 3.72 Å and 3.47 Å, respectively. Further coordination is realized with the N atom of the pyrrolidine ring of Pro205 at a distance of 3.64 Å and an O atom of bound hexanediol at 3.68 Å distance (Figure 3b).

The substitutions Ile44Leu and Gln102His introduced into the structure of DbeA result in the elimination of the second halide-binding site (Figure 3c). The site is occupied by a water molecule in the mutant enzyme. Apparently, insufficient space is left for the positioning of the second halide anion as observed in the DbeAwt structure. The atoms of three residues: Thr40 O$^{\gamma 1}$, His102 N$^{\delta 1}$ and Gly37 N, with hydrogen bond distances of 2.74 Å, 2.79 Å, and 2.94 Å, respectively, coordinated water molecule W71 that was modelled instead of the halide anion (Figure 3c). Further coordination is realized by water molecule W11 at a distance of 2.73 Å. Water molecule W11 is situated between two halide-binding sites, 6.73 Å away from the iodide ion in the canonical active site of the protein. Water W11 is stabilized by interaction with the O$^{\delta 2}$ atom of catalytic nucleophile Asp103 at 2.71 Å distance.

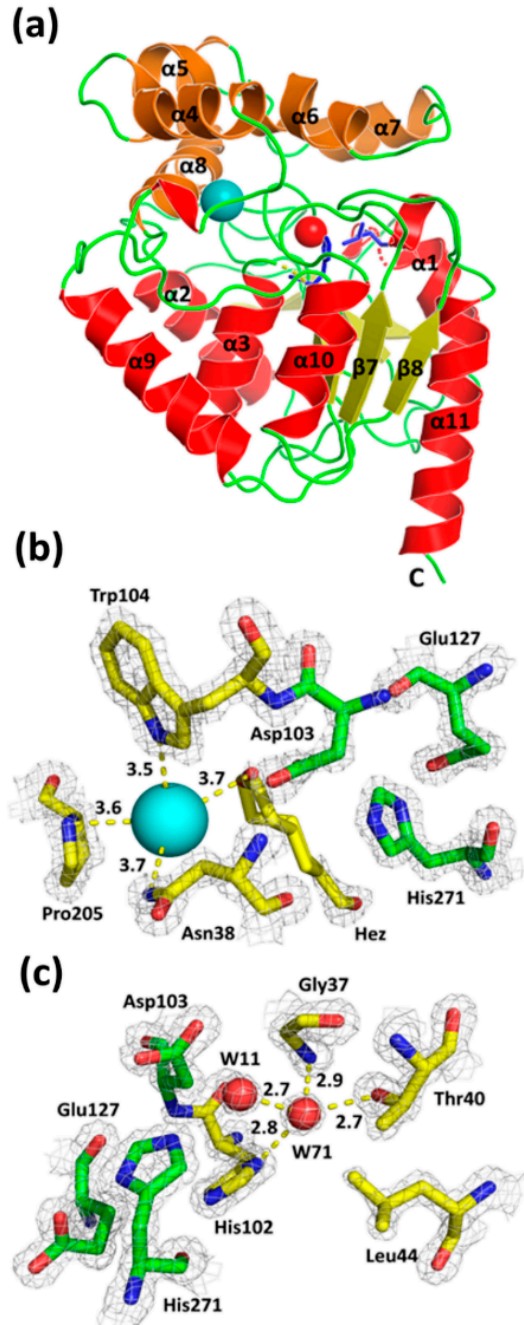

**Figure 3.** The overall structure of DbeAΔCl (**a**), close-up view of the canonical DbeAΔCl active site (**b**) and second halide-binding site (**c**). The 2F$_O$-F$_c$ electron-density map contoured at 2σ is shown in grey (**a**) Cα ribbon trace shows elements of the protein secondary structure. The α-helices are coloured red for the main domain and brown for the cap domain; β-strands are coloured yellow; loops are shown in green; iodide ion is presented as a cyan sphere, water molecule (W71) is shown as a red sphere; the two point substitutions Ile44Leu + Gln102His introduced into DbeA are highlighted as blue sticks. (**b**) The iodide ion in the active site is presented as a cyan sphere with coordination interaction distances in Å and highlighted by yellow dashed lines; hexanediol (Hez) (shown in two alternative conformations) and amino acids Asp103, Trp104, Pro205 coordinating the iodide ion are shown as sticks with carbon atoms coloured yellow. Carbon atoms of catalytic triad are highlighted in green. (**c**) The water molecules W11 and W71 are presented as red spheres; amino acids coordinating water molecule W71 are shown as sticks with carbon atoms coloured in yellow with and interactions with distances in Å shown by yellow dashed lines. Water molecule W71 is located in second-halide binding site.

The sequence of DbeAΔCl from *B. elkanii* (PDB ID code: 6s42) was aligned (Figure 4) and compared with five known sequences of HLDs deposited in the PDB: DbeA wild type from *B. elkanii* (PDB ID code: 4k2a [7]), DhaA from *Rhodococcus species* (PDB ID code: 1bn6 [1]), DhlA from *Xanthobacter autotrophicus* (PDB ID code: 1cij [28]), LinB from *Sphingomonas paucimobilis* (PDB ID code: 1cv2 [29]) and DmbA from *Mycobacterium tuberculosis* (PDB ID code: 2qvb [30]). The reason for selection of these dehalogenases is that DbeAwt is the same protein without two point mutations occurring in DbeAΔCl with 99.3% sequence identity, DhaA as HLD with highest sequence identity shows 50.7% and DhlA as HLD with lowest sequence identity displays 26.5% in comparison to DbeAΔCl. Alignment of DbeAΔCl with different kind of haloalkane dehalogenase LinB from the same substrate specificity group (SSG-I) demonstrates 47.4% sequence identity similar to DmbA as another type of HLD from different SSG-III (44.8%) [31].

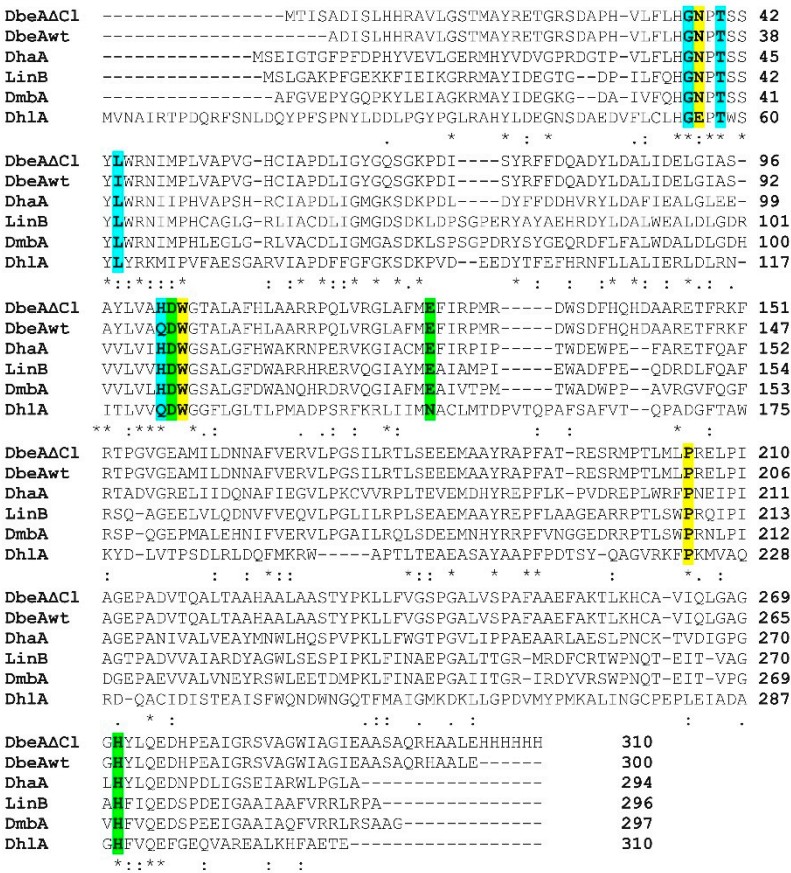

**Figure 4.** Multiple sequence alignment of DbeAΔCl with five Haloalkane dehalogenases (HLDs) deposited in PDB. Amino acids from canonical HLD active site are highlighted in yellow, catalytic triad residues are highlighted in green and second halide binding site residues are highlighted in cyan. Sequence alignment was performed by ClustalW [32].

The molecular structures of DbeA, DhaA, LinB, DmbA, and DhlA were superposed with the DbeAΔCl enzyme's Cα atoms with root mean square deviations of 0.335, 1.018, 0.985, 1.127 and 2.112 Å, respectively. 3D-superpositions of the DbeAΔCl structure with the closest similarity model (DhaA) and the lowest identity model (DhlA) are shown in Figure 5a.

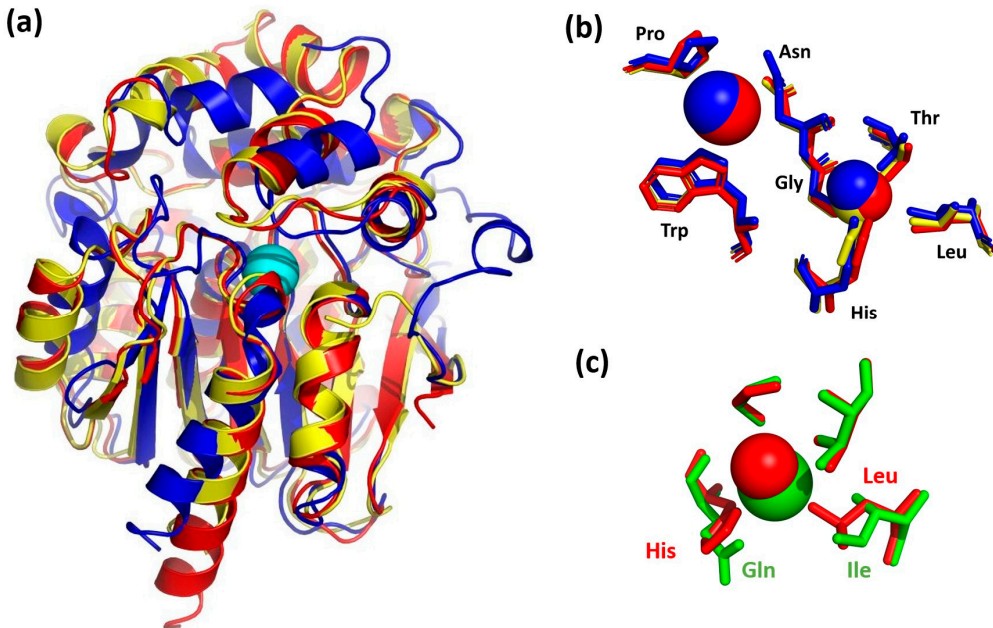

**Figure 5.** Structural comparison of DbeAΔCl with homologous dehalogenases. (**a**) DbeAΔCl secondary structure superposition with DhaA and DhlA. Cα ribbon trace shows elements of the protein secondary structures. The DbeAΔCl is coloured in red; DhaA is coloured in blue; DhlA is shown in yellow; iodide ion (cyan sphere) is placed in the canonical active site of DbeAΔCl. (**b**) Superposition of DbeAΔCl, DhaA and DhlA active sites. Amino acids of DbeAΔCl are shown as red sticks; amino acids of DhaA are shown as blue sticks; amino acids of DhlA are shown as yellow sticks. The iodide ion and water molecule W11 bound to DbeAΔCl are represented as red spheres; bromide ion and water are shown as blue spheres for the DhaA structure and the single water molecule in the DhlA is shown as yellow sphere. (c) Superposition of second halide binding site residues of mutant and wild type DbeA. DbeAΔCl amino acids are shown as red sticks; amino acids of DbeAwt are shown as green sticks. Water W71 is shown as red sphere and Cl⁻ ion is shown as green sphere.

In general, the secondary structure elements of the core domain are better conserved compared to the cap domain for the most dehalogenases. Significant differences in cap domains define substrate specificity and variability [1]. The considerable divergence among the superimposed structures was also observable at the relatively disordered N- and C-terminal parts of the proteins (Figure 5a). The position of the active site residues was well conserved among HLDs with some differences in DhlA structure. DbeAΔCl, DbeA, LinB, DhaA and DmbA belong to the HLD-II subfamily with Asp-His-Glu catalytic triad and Asn-Trp halide binding residues [4] whereas DhlA belongs to the HLD-I subfamily with an Asp-His-Asp + Trp-Trp catalytic pentad composition. The positions of the halide-binding residues Asn38 and Trp104 in the DbeAΔCl structure is conserved in comparison to the rest of HLD structures. The position of the catalytic triad (Asp103; His271 and Glu127) is well conserved among the dehalogenases from HLD-II subfamily. The side chain of the catalytic acid Asp in DhlA is situated slightly deeper in the active site cavity in comparison to HLD-II structures.

Superposition of the iodide ion at the DbeAΔCl active site with halide ions in the vicinity of halide-stabilizing residues of the rest structures reveals some structural differences. DhlA and DmbA contain halide ions inside the active site: Br⁻ (DhlA) and Cl⁻ (DmbA) with shifts in 0.47 and 0.39 Å from the DbeAΔCl iodide ion position, respectively. However, there is only a water molecule in the LinB structure (0.27 Å away from iodide ion of DbeAΔCl) and an empty space in the DhaA structure (the closest water molecule is located 6.48 Å away from the iodide ion position).

Water molecule W71, which substitutes the halide ion at the second halide-binding site of the DbeA wild type structure, was found at the canonical place where only a water molecule is present in all the HLDs with a shift of 1.39 Å compared to the DhaA structure, 0.80 Å to the DhlA, 0.55 Å to the

LinB and 0.43 Å to the DmbA, and 1.28 Å away compared to the second chloride anion in the DbeA structure. Superposition of halide binding sites of the DbeAΔCl structure with halide binding sites of the closest relative (DhaA) and the lowest identity HLD (DhlA) is shown in the Figure 5b.

## 4. Conclusions

After many crystallization experiments and optimization cycles, we found that two point mutations deeply buried in the structure have a significant influence on the crystallization of the macromolecule. Finally, random microseeding experiments with a seed stock prepared from small 3D crystals obtained at a protein concentration of 20 mg.ml$^{-1}$ and 0.2 M sodium iodide plus 20% (w/v) PEG 3350 as the precipitant solutions produced crystals of the double mutant DbeAΔCl from *B. elkanii* USDA94. These crystals were of sufficient quality for X-ray diffraction experiments. The structure of DbeAΔCl was solved using molecular replacement and refined to 1.4 Å resolution. Overall, the structure is very similar to other HLDs structures of the α/β hydrolase fold superfamily (EC 3.8.1.5). The substitutions Ile44Leu and Gln102His resulted in a space reduction of the second halide-binding site and thus incapability of DbeAΔCl to bind a second halide anion as compared to the wild type structure. Instead of a chloride anion, a water molecule (W71) was found in the site of which the consequences are, DbeAΔCl is less active and less stable in the presence of chloride salts when compared with the DbeAwt enzyme [7]. The DbeAΔCl structure of B. *elkanii* was aligned and compared with five molecular structures of haloalkane dehalogenases selected from the PDB. The elements of the secondary structure and the catalytic pentad are well conserved. Superposition of the iodide ion at the DbeAΔCl active site with the other structures reveals some structural differences. The water molecule W71 located at the compromised second halide-binding site of DbeAΔCl was coordinated at the canonical place as compared to other HLDs.

**Author Contributions:** T.P., J.R.M. and I.K.S. designed the experiments and solved the structure. T.P., B.K., P.G., J.R.M. and M.K. analyzed the data. T.P. and B.K. wrote the manuscript. R.C., J.D., A.M., P.H. and A.S. provided technical support.

**Funding:** This work was supported by the Grant Agency of the Czech Republic 17-24321S; DAAD mobility grant DAAD-16-09; ERDF project CZ.02.1.01/0.0/0.0/15_003/0000441; Ministry of Education, Youth and Sports of the Czech Republic (CZ.1.05/2.1.00/01.0024, CZ.1.05/2.1.00/01.0001, and LM2015055); GAJU 17/2019/P.

**Acknowledgments:** The diffraction data were collected on the beam line MX14.1 at the BESSY II electron storage ring operated by the Helmholtz-Zentrum Berlin. We would particularly like to acknowledge the help and support of Manfred S. Weiss during data collection. Also, we would like to thank Stefan A. Kolek (Douglas Instruments Ltd, Hungerford, UK) for providing crucial information about the random seeding experiment during the 2014 FEBS-Instruct practical course PC14-005 at Nove Hrady, Czech Republic.

**Conflicts of Interest:** No interest conflict exists among the authors.

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
