# Peer review of "Crystallization and Crystallographic Analysis of a Bradyrhizobium Elkanii USDA94 Haloalkane Dehalogenase Variant with an Eliminated Halide-Binding Site"

_crystals, doi:10.3390/cryst9070375_

Round 1
Reviewer 1 Report
My review report is attached.

Author Response
The manuscript contains some minor problems which have to be addressed that will improve quality of the manuscript and its absorption by many readers. These problems are enumerated below:
1. All reproducible results should be reported in the present tense because they are time independent. This also applies to narration of universal facts and observations. For example, Mr X found that Earth is round. In the case of this work, for example, crystals belong to the monoclinic space group, not belonged.
In response to the reviewer’s suggestions, we have made a few corrections in the manuscript (e.i. page 1 line 30, etc.).
2. Page 4, line 176: How do you know that these are quasicrystals. Was this checked?
We did not check the crystals from Figure 1a for the quasicrystals nature. We renamed the term of “quasicrystals” to “microcrystals” what corresponds to the reality better.
3. Page 5, line 191: "these crystals were not suitable for X-ray data collection experiments". Why not? For the poor quality of crystals?
To solve the crystal structure the diffraction to 8 Å maximum resolution is not enough. The manuscript was supplemented with the text on page 5 lines 197-199:
The quality of these crystals did not allow to record good diffraction data and additional strategies were needed to improve the size and shape of obtained crystals.
4. Fig 1b: Figure 1b is completely illegible, you can hardly see these needles. You need to improve the contrast.
The contrast was increased and Figure 1b was replaced by the new figure in the manuscript.
5. Page 5, line 197: didn't → did not.
The corrections were done.
Please find all the provided correction in track change mode.

Reviewer 2 Report
The central argument of this manuscript is that the crystal structure of DbeAΔCl structurally confirms that the 2nd halide site is accommodated by Ile44 and Gln102, and that reverting them back to the more common residues seen in related HLDs abrogates halide binding here. Towards this, some suggestions are offered to strengthen the author's arguments.
A brief mention in the text about any other structural differences between DbeA and other HLDs would be useful. Are they exactly identical with the exception of the additional halide binding site? How about RMSD? RMSD comparison between the current structure with WT DbeA would be informative as well.
A figure showing a superposition of several HLD structures with an emphasis on the 2nd Halide binding site to illustrate that there is indeed a difference in pocket size would be very helpful (similar to what was done for reference 7, Chaloupkova et al., 2014). Figure 3c illustrates the current structure, but without a point of comparison with DbeA-WT or other single pocket halide-binding structures it is difficult to appreciate this difference in cavity size.
Along these lines it would also be helpful to analyze the distances between the relevant residues and the chloride ion (for DbeA) or water molecule for (DbeAΔCl), especially at such an impressive resolution. This is done in figure 3c, but there is no comparison with that from other structures. Furthermore, if the DbeA-WT and DbeAΔCl models were superimposed, what would the hypothetical distance be between DbeAΔCl Leu44 and His102 and DbeA-WT chloride? Would this be sterically unfeasible? Assuming this is true, illustrating such would greatly strengthen the results here. A table that compares these distances along with a graph could also be helpful.
A figure with an alignment of various HLD sequences around the halide binding sites would be very useful to allow readers to judge the naturally occurring frequency of residue substitutions between different HLD proteins.
Author Response
The central argument of this manuscript is that the crystal structure of DbeAΔCl structurally confirms that the 2nd halide site is accommodated by Ile44 and Gln102, and that reverting them back to the more common residues seen in related HLDs abrogates halide binding here. Towards this, some suggestions are offered to strengthen the author's arguments.
A brief mention in the text about any other structural differences between DbeA and other HLDs would be useful. Are they exactly identical with the exception of the additional halide binding site? How about RMSD? RMSD comparison between the current structure with WT DbeA would be informative as well.
According to the reviewer’s suggestions, the following text was added to the manuscript:
The sequence of DbeAΔCl from B. elkanii (PDB ID code: 6s42) was aligned (Figure 4) and compared with five known sequences of HLDs deposited in the PDB: DbeA wild type from B. elkanii (PDB ID code: 4k2a [7]), DhaA from Rhodococcus species (PDB ID code: 1bn6 [1]), DhlA from Xanthobacter autotrophicus (PDB ID code: 1cij [28]), LinB from Sphingomonas paucimobilis (PDB ID code: 1cv2 [29]) and DmbA from Mycobacterium tuberculosis (PDB ID code: 2qvb [30]). The reason for selection of these dehalogenases is that DbeAwt shows 99.3% sequence identity, DhaA as HLD with highest sequence identity shows 50.7% and DhlA as HLD with lowest sequence identity displays 26.5% in comparison to DbeAΔCl. Alignment of DbeAΔCl with different kind of haloalkane dehalogenase LinB from the same substrate specificity group (SSG-I) demonstrates 47.4% sequence identity similar to DmbA as another type of HLD from different SSG-III (44.8%) [31].
The molecular structures of DbeA, DhaA, LinB, DmbA, and DhlA were superposed with the DbeAΔCl enzyme’s Cα atoms with root mean square deviations of 0.335, 1.018, 0.985, 1.127 and 2.112 Å, respectively. 3D-superpositions of the DbeAΔCl structure with the closest similarity model (DhaA) and the lowest identity model (DhlA) are shown in Figure 5a.
In general, the secondary structure elements of the core domain are better conserved compared to the cap domain for the most dehalogenases. Significant differences in cap domains define substrate specificity and variability [1]. The considerable divergence among the superimposed structures was also observable at the relatively disordered N- and C-terminal parts of the proteins (Figure 5a). The position of the active site residues was well conserved among HLDs with some differences in DhlA structure. DbeAΔCl, DbeA, LinB, DhaA and DmbA belong to the HLD-II subfamily with Asp-His-Glu catalytic triad and Asn-Trp halide binding residues [4] whereas DhlA belongs to the HLD-I subfamily with an Asp-His-Asp + Trp-Trp catalytic pentad composition. The positions of the halide-binding residues Asn38 and Trp104 in the DbeAΔCl structure is conserved in comparison to the rest of HLD structures. The position of the catalytic triad (Asp103; His271 and Glu127) is well conserved among the dehalogenases from HLD-II subfamily. The side chain of the catalytic acid Asp in DhlA is situated slightly deeper in the active site cavity in comparison to HLD-II structures.
A figure showing a superposition of several HLD structures with an emphasis on the 2nd Halide binding site to illustrate that there is indeed a difference in pocket size would be very helpful (similar to what was done for reference 7, Chaloupkova et al., 2014). Figure 3c illustrates the current structure, but without a point of comparison with DbeA-WT or other single pocket halide-binding structures it is difficult to appreciate this difference in cavity size.
Figure 5 illustrating the superposition of DbeAΔCl 3D structure and active sites with other HLDs was created and added to the manuscript to the page 9.
Along these lines it would also be helpful to analyze the distances between the relevant residues and the chloride ion (for DbeA) or water molecule for (DbeAΔCl), especially at such an impressive resolution. This is done in figure 3c, but there is no comparison with that from other structures. Furthermore, if the DbeA-WT and DbeAΔCl models were superimposed, what would the hypothetical distance be between DbeAΔCl Leu44 and His102 and DbeA-WT chloride? Would this be sterically unfeasible? Assuming this is true, illustrating such would greatly strengthen the results here. A table that compares these distances along with a graph could also be helpful.
To improve the strength of the paper the following text and Figure 5c were implemented to the manuscript:
Superposition of the iodide ion at the DbeAΔCl active site with halide ions in the vicinity of halide-stabilizing residues of the rest structures reveals some structural differences. DhlA and DmbA contain halide ions inside the active site: Br- (DhlA) and Cl- (DmbA) with shifts in 0.47 and 0.39 Å from the DbeAΔCl iodide ion position, respectively. However, there is only a water molecule in the LinB structure (0.27 Å away from iodide ion of DbeAΔCl) and an empty space in the DhaA structure (the closest water molecule is located 6.48 Å away from the iodide ion position).
Water molecule W71, which substitutes the halide ion at the second halide-binding site of the DbeA wild type structure, was found at the canonical place where only a water molecule is present in all the HLDs with a shift of 1.39 Å compared to the DhaA structure, 0.80 Å to the DhlA, 0.55 Å to the LinB and 0.43 Å to the DmbA, and 1.28 Å away compared to the second chloride anion in the DbeA structure. Superposition of halide binding sites of the DbeAΔCl structure with halide binding sites of the closest relative (DhaA) and the lowest identity HLD (DhlA) is shown in the Figure 5b.
A figure with an alignment of various HLD sequences around the halide binding sites would be very useful to allow readers to judge the naturally occurring frequency of residue substitutions between different HLD proteins.
Figure 4 was prepared and added to the manuscript to the page 9 to characterize the alignment of DbeAΔCl with five HLDs selected from the PDB.
Finally, the abstract was extended about sentence below:
The DbeAΔCl molecular structure was characterized and compared with five known haloalkane dehalogenases selected from the Protein Data Bank.
Conclusions part was extended about text below:
The DbeAΔCl structure of B. elkanii was aligned and compared with five molecular structures of haloalkane dehalogenases deposited in the PDB. The elements of the secondary structure and the catalytic pentad are well conserved. Superposition of the iodide ion at the DbeAΔCl active site with the other structures reveals some structural differences. The water molecule W71 located at the compromised second halide-binding site of DbeAΔCl was coordinated at the canonical place as compared to other HLDs.
Five more references were added to the manuscript:
1. Pikkemaat, M.G.; Ridder, I.S.; Rozeboom, H.J.; Kalk, K.H.; Dijkstra, B.W.; Janssen, D.B. Crystallographic and kinetic evidence of a collision complex formed during halide import in haloalkane dehalogenase. Biochemistry. 1999, 38, 12052–12061. [DOI: 10.1021/bi990849w]
2. Marek, J.; Vevodova, J.; Smatanova, I.K.; Nagata, Y.; Svensson, L.A.; Newman, J.; Takagi, M.; Damborsky, J. Crystal structure of the haloalkane dehalogenase from Sphingomonas paucimobilis UT26. Biochemistry. 2000, 39, 14082–14086. [DOI: 10.1021/bi001539c]
3. Mazumdar, P.A.; Hulecki, J.C.; Cherney, M.M.; Garen, C. R.; James, M.N.G. X-ray crystal structure of Mycobacterium tuberculosis haloalkane dehalogenase Rv2579. Biochi Biophys Acta. 2008, 1784, 351–362. [DOI: 10.1016/j.bbapap.2007.10.014]
4. Koudelakova T.; Bidmanova S.; Dvorak P.; Pavelka A.; Chaloupkova R.; Prokop Z.; Damborsky J. Haloalkane dehalogenases: biotechnological applications. Biotechnol J. 2013, 8(1), 32-45. [DOI: 10.1002/biot.201100486]
5. Larkin M.A.; Blackshields G.; Brown N.P.; Chenna R.; McGettigan P.A.; McWilliam H.; Valentin F.; Wallace I.M.; Wilm A.; Lopez R.; Thompson J.D.; Gibson T.J; Higgins D.G. Clustal W and Clustal X version 2.0. Bioinformatics. 2007, 23, 2947-2948. [DOI: 10.1093/bioinformatics/btm404]
According to all the recommendations the manuscript name was changed from “Crystallization and crystallographic characterization of a Bradyrhizobium elkanii USDA94 haloalkane dehalogenase variant with an eliminated halide-binding site” to
“Crystallization and crystallographic analysis of a Bradyrhizobium elkanii USDA94 haloalkane dehalogenase variant with an eliminated halide-binding site”
Please find all the provided correction in track change mode.
